# Toward Pathogenic Biofilm Suppressors: Synthesis of Amino Derivatives of Pillar[5]arene and Supramolecular Assembly with DNA

**DOI:** 10.3390/pharmaceutics15020476

**Published:** 2023-01-31

**Authors:** Yulia I. Aleksandrova, Dmitriy N. Shurpik, Viktoriya A. Nazmutdinova, Olga A. Mostovaya, Evgenia V. Subakaeva, Evgenia A. Sokolova, Pavel V. Zelenikhin, Ivan I. Stoikov

**Affiliations:** 1A.M. Butlerov Chemistry Institute, Kazan Federal University, Kremlevskaya, 18, 420008 Kazan, Russia; 2Institute of Fundamental Medicine and Biology, Kazan Federal University, Kremlevskaya, 18, 420008 Kazan, Russia

**Keywords:** water-soluble pillar[5]arene, salmon sperm DNA, DNA compactification, gene transfection vector, antibacterial systems, biofilms, resistance, *St. Aureus*

## Abstract

New amino derivatives of pillar[5]arene were obtained in three stages with good yields. It was shown that pillar[5]arene containing thiaether and tertiary amino groups formed supramolecular complexes with low molecular weight model DNA. Pillar[5]arene formed complexes with a DNA nucleotide pair at a ratio of 1:2 (macrocycle/DNA base pairs), as demonstrated by UV-visible and fluorescence spectroscopy. The association constants of pillar[5]arene with DNA were lgKass1:1 = 2.38 and lgKass1:2 = 5.07, accordingly. By using dynamic light scattering and transmission electron microscopy, it was established that the interaction of pillar[5]arene containing thiaether and tertiary amino groups (concentration of 10−5 M) with a model nucleic acid led to the formation of stable nanosized macrocycle/DNA associates with an average particle size of 220 nm. It was shown that the obtained compounds did not exhibit a pronounced toxicity toward human adenocarcinoma cells (A549) and bovine lung epithelial cells (LECs). The hypothesis about a possible usage of the synthesized macrocycle for the aggregation of extracellular bacterial DNA in a biofilm matrix was confirmed by the example of *St. Aureus*. It was found that pillar[5]arene at a concentration of 10^−5^ M was able to reduce the thickness of the *St. Aureus* biofilm by 15%.

## 1. Introduction

A serious problem in the fight against infectious diseases is the ability of pathogenic microorganisms to form biofilms, which increases their resistance to various antibiotics [1,2,3,4,5]. About 65–80% of known bacterial infections are associated with the development of biofilms [4]. One of the reasons for the tolerance of biofilm microorganisms to an antibiotic is the complex multicomponent structure of a self-producing matrix that prevents drug diffusion [6,7]. Several enzymes, being part of the matrix, are capable of inactivating the antibiotic, resulting in a special effect on resistance as well as extracellular DNA, which, due to a transformation, facilitates the spread of antibiotic resistance genes [3]. Extracellular DNA plays an important role in biofilm development; it is involved in the attachment of the biofilm to the substrate and has the ability to chelate magnesium, creating a resistance to the influence of antimicrobial peptides and inhibiting the transport of antibiotics [8].

One of the modern methods of combating pathogenic biofilms is the use of DNA-binding proteins [9]. Among the DNA-binding proteins systems, DNases are the most well-studied. These DNases have an antibiofilm activity, provoking the degradation of extracellular DNA [10]. For example, streptococcal DNase streptodornase (a component of a therapeutic agent for the treatment of purulent diseases) inhibits the formation of *P. Aeruginosa* biofilms. The advantages of DNases in the fight against pathogenic microorganisms include their low toxicity and high efficiency. The use of DNA-binding proteins increases the effectiveness of antibiotics in the fight against pathogenic microorganisms [11]. However, all these systems have significant drawbacks such as a low stability and a high cost of the final biological preparations.

These limitations can be overcome by using synthetic biomimetic analogs of natural DNA-binding proteins. Compounds based on macrocyclic systems can be used as DNA-binding agents [12,13]. Thus, the problem of suppressing the development of pathogenic biofilms using macrocycles can be solved by creating nanosystems capable of selectively binding bacterial extracellular DNA. Additionally, such macrocyclic compounds may have antimicrobial properties [14,15] or be effective additives to antibiotics to increase the effectiveness of combination therapies for bacterial diseases.

Due to their low toxicity, water solubility and high specificity with respect to biological substrates, pillar[n]arenes are unique compounds [16,17,18,19] that can be macrocyclic structures for binding extracellular DNA, inhibiting the growth and development of pathogenic biofilms. It should be noted that DNA effectively interacts with cationic amphiphiles. This type of molecule can be used to disrupt the interaction between extracellular DNA and other components of the exopolymeric matrix from a bacterial biofilm [20]. Numerous examples of effective DNA binding by pillararenes are described in the literature [21,22]. For instance, in the works of Evtugyn [23], Xin Ran [24] and Yanli Zhao [25], the pillararene macrocyclic platform acted as a DNA sensor. In our scientific group, delivery visualization systems based on the nanoassociates of pillar [5]arene, DNA and dye have been studied [26]. The work of Wang [27] reported a nanosystem for the targeted delivery of an anticancer drug (doxorubicin) on the pillararene platform with a DNA-binding ability. However, many delivery systems based on macrocycles (including pillararenes) are non-specific and tend to form complexes with metal cations [28], which limits their usage owing to the presence of a large number of micro- and macroelements in the biofilm matrix.

In this research, we propose a new approach to fight pathogenic biofilms through the formation of supramolecular macrocycle/extracellular DNA associates. We propose the use of non-toxic, water-soluble derivatives of pillar[5]arene as a new generation of DNA-binding agent.

## 2. Materials and Methods

Detailed information on the equipment, methods and physical–chemical characterization is presented in the Appendix A.

### General Procedure for the Synthesis of Macrocycles ***3***–***6***

In a round-bottomed flask equipped with a magnetic stirrer, 0.20 g (0.0965 mmol) of macrocycle **2** was dissolved in 4 mL of toluene; 1.93 mmol of amine (0.1 mL of hydrazine hydrate, 0.27 mL of N,N-diethylaminoethylamine, 0.21 mL of N,N-dimethylaminoethylamine or 0.25 mL of morpholine) was then added. To the reaction mixture, 4 mL of methanol was added. The reaction was carried out at 150 °C for 50 h in an argon atmosphere. The reaction mixture was then evaporated on a rotary evaporator, dissolved in 15 mL of methylene chloride and washed with distilled water (3 × 30 mL). The organic phase was then separated and evaporated on a rotary evaporator.

4,8,14,18,23,26,28,31,32,35-deca-((hydrazidocarbonil-2-sulfanedyl)ethoxy)-pillar[5]arene (**3**)

Yield: 0.11 g (60%).

^1^H NMR (DMSO-*d_6_*): 2.37 t (20H; ^3^*J*_HH_ = 7.0 Hz; -CH_2_C(O)), 2.82 t (20H; ^3^*J*_HH_ = 6.9 Hz; -SCH_2_-), 2.88–3.05 m (20H; -CH_2_S-), 3.57 s (10H; -CH_2_-), 4.04–4.32 m (20H; -OCH_2_-), 6.91 s (10H; ArH) and 9.07 s (NH).

^13^C NMR (DMSO-*d_6_*): 26.84, 31.08, 34.36, 67.68, 114.48, 128.07, 149.82 and 175.01.

IR (v; sm^−1^): 3258 (-NH-; -NH_2_), 2923 (-C_Ph_-H), 2864 (-CH_2_-; C_Ph_-O-CH_2_), 1654 (-C=O Amide I), 1600 (-NH- Amide II), 1496 (-CH_2_-), 1465 (C_Ph_-C_Ph_), 1249 (Amide III), 1201 (C_Ph_-O-CH_2_), 1100 (C_Ph_-O-CH_2_), 1063 (C_Ph_-O-CH_2_), 1022 (-C_Ph_-H) and 702 (C-S).

MS (MALDI-TOF): calc. [M^+^ + Na^+^] *m*/*z* = 2095.7; found [M^+^ + Na^+^] *m*/*z* = 2095.8.

Found (%): C, 49.45; H, 6.25; N, 13.40; O, 15.37; S, 15.53. Calc. for C_85_H_130_N_20_O_20_S_10_ (%): C, 49.26; H, 6.32; N, 13.52; O, 15.44; S, 15.47.

4,8,14,18,23,26,28,31,32,35-deca-((2-(N,N-diethyl)ethylcarbamoyl-2-sulfanedyl)ethoxy)-pillar[5]arene (***4***)

Yield: 0.24 g (87%)

^1^H NMR (DMSO-*d_6_*): 0.92 t (60H; ^3^*J*_HH_ = 6.9 Hz; -CH_3_), 2.32–2.46 m (60H; -NHCH_2_CH_2_N; -CH_2_CH_3_), 2.66 t (20H; ^3^*J*_HH_ = 7.2 Hz; -CH_2_C(O)), 2.82 t (20H; ^3^*J*_HH_ = 6.7 Hz; -SCH_2_-), 2.96 t (20H; ^3^*J*_HH_ = 6.8 Hz; -CH_2_S-), 3.57–3.63 m (20H; -NHCH_2_CH_2_N-), 3.67 s (10H; -CH_2_-), 3.87 and 4.22 br. AB-system (20H; -OCH_2_-), 6.89 s (10H; ArH) and 7.81 s (NH).

^13^C NMR (DMSO-*d_6_*): 11.74; 26.84; 27.77; 31.07; 34.37; 36.03; 46.65; 51.40; 62.02; 114.37; 128.11; 148.92; 171.92.

IR (v; sm^−1^): 3298 (N-H), 2925 (-C_Ph_-H), 2868 (-CH_2_-; C_Ph_-O-CH_2_), 1648 (-C=O Amide I), 1543 (-NH- Amide II), 1497 (-CH_2_-), 1465 (C_Ph_-C_Ph_), 1249 (Amide III), 1199 (C_Ph_-O-CH_2_), 1092 (C_Ph_-O-CH_2_), 1064 (C_Ph_-O-CH_2_), 1023 (-C_Ph_-H) and 703 (C-S).

MS (MALDI-TOF): calc. [M^+^ + 11K^+^ + 4H^+^] *m*/*z* = 3346.4, [M^+^-5(C_2_H_5_)_2_NCH_2_CH_2_C(O)CH_2_CH_2_ + K^+^] *m*/*z* = 2095.1 and [(C_2_H_5_)_2_NCH_2_CH_2_C(O)CH_2_CH_2_ + 5H^+^] *m*/*z* = 176.7; found [M^+^ + 11K^+^ + 4H^+^] *m*/*z* = 3346.4, [M^+^-5(C_2_H_5_)_2_NCH_2_CH_2_C(O)CH_2_CH_2_ + K^+^] *m*/*z* = 2095.5 and [(C_2_H_5_)_2_NCH_2_CH_2_C(O)CHCH_2_ + 5H^+^] *m*/*z* = 176.9

Found (%): C, 59.91; H, 8.68; N, 9.65; O, 10.84; S, 10.92. Calc. for C_145_H_250_N_20_O_20_S_10_ (%): C, 59.76; H, 8.65; N, 9.61; O, 10.98; S, 11.00.

4,8,14,18,23,26,28,31,32,35-deca-((2-(N,N-dimethyl)ethylcarbamoyl-2-sulfanedyl)ethoxy)-pillar[5]arene (***5***)

Yield: 0.23 g (89%)

^1^H NMR (DMSO-*d_6_*): 2.14 s (60H; -CH_3_), 2.41 t (20H; ^3^*J*_HH_ = 6.3 Hz; -NHCH_2_CH_2_N), 2.66 t (20H; ^3^*J*_HH_ = 7.0 Hz; -CH_2_C(O)), 2.84 t (20H; ^3^*J*_HH_ = 7.1 Hz; -SCH_2_-), 2.96 t (20H; ^3^*J*_HH_ = 5.2 Hz; -CH_2_S-), 3.57–3.63 m (20H; -NHCH_2_CH_2_N-), 3.67 s (10H; -CH_2_-), 3.88 and 4.22 br. AB-system (20H; -OCH_2_-) and 6.89 s (10H; ArH).

^13^C NMR (DMSO-*d_6_*): 31.23, 33.93, 34.40, 35.55, 42.29, 51.46, 55.74, 67.76, 114.44, 128.19, 148.98 and 171.99.

IR (v; sm^−1^): 3283 (N-H), 2926 (-C_Ph_-H), 2862 (-CH_2_-, C_Ph_-O-CH_2_), 1663 (-C=O Amide I), 1548 (-NH- Amide II), 1497 (-CH_2_-), 1464 (C_Ph_-C_Ph_), 1373 (-N-tert), 1249 (Amide III), 1203 (C_Ph_-O-CH_2_), 1097 (C_Ph_-O-CH_2_), 1055 (C_Ph_-O-CH_2_), 1025 (-C_Ph_-H) and 705 (C-S).

MS (MALDI-TOF): calc. [M^+^ + Na^+^ + 4K^+^] *m*/*z* = 2836.15 and [M^+^-5(CH_3_)_2_NCH_2_CH_2_C(O)CH_2_CH_2_ + 4H^+^] *m*/*z* = 1924.7; found [M^+^ + Na^+^ + 4K^+^] *m*/*z* = 2835.3 and [M^+^-5(CH_3_)_2_NCH_2_CH_2_C(O)CH_2_CH_2_ + 4H^+^] *m*/*z* = 1924.5.

Found (%): C, 57.11; H, 8.10; N, 10.58; O, 12.02; S, 12.19. Calc. for C_125_H_210_N_20_O_20_S_10_ (%): C, 57.00; H, 8.04; N, 10.64; O, 12.15; S, 12.17.

4,8,14,18,23,26,28,31,32,35-deca-((2-(morfolin)ethylcarbamoyl-2-sulfanedyl)ethoxy)-pillar[5]arene (***6***)

Yield: 0.22 g (75%)

^1^H NMR (DMSO-*d_6_*): 2.22–2.45 m (60H; -NHCH_2_CH_2_N; -NCH_2__morph_.), 2.66 t (20H; ^3^*J*_HH_ = 7.0 Hz; -CH_2_C(O)), 2.84 t (20H; ^3^*J*_HH_ = 6.9 Hz; -SCH_2_-), 2.96 br.s (20H; -CH_2_S-), 3.54–3.63 m (60H; -NHCH_2_CH_2_N-; -CH_2_O_morph_.), 3.67 s (10H; -CH_2_-), 3.89 and 4.21 br. AB-system (20H; -OCH_2_-) and 6.89 s (10H; ArH).

^13^C NMR (DMSO-*d_6_*): 26.18; 26.88; 31.11; 34.40; 35.93; 51.46; 53.32; 66.19; 67.65; 114.41; 128.27; 148.96; 172.00.

IR (v; sm^−1^): 3384 (N-H), 2948 (-N-tert), 2924 (-C_Ph_-H), 2863 (-CH_2_-; C_Ph_-O-CH_2_), 1655 (-C=O Amide I), 1558 (-NH- Amide II), 1497 (-CH_2_-), 1463 (C_Ph_-C_Ph_), 1358 (-N-tert), 1248 (Amide III), 1202 (C_Ph_-O-CH_2_), 1174 (-C-O-C-_morph_), 1115 (C_Ph_-O-CH_2_), 1066 (C_Ph_-O-CH_2_), 1025 (-C_Ph_-H) and 703 (C-S).

MS (MALDI-TOF): calc. [M^+^ + 6K^+^ + 3Na^+^] *m*/*z* = 3557.2 and [M^+^-6(C_2_H_4_O)NCH_2_CH_2_C(O)CHCH_2_ + 4K^+^] *m*/*z* = 2096.66; found [M^+^ + 6K^+^ + 3Na^+^] *m*/*z* = 3556.6 and [M^+^-5(C_2_H_4_O)NCH_2_CH_2_C(O)CHCH_2_ + 4K^+^] *m*/*z* = 2096.6.

Found (%): C, 57.00; H, 7.73; N, 9.31; O, 15.58; S, 10.38. Calc. for C_145_H_230_N_20_O_30_S_10_ (%): C, 57.02; H, 7.59; N, 9.17; O, 15.72; S, 10.50.

## 3. Results and Discussion

The proposed non-toxic pillar[5]arene platform with DNA-binding characteristics has the potential to develop systems for inhibiting the growth of pathogenic microorganisms. Macrocyclic systems containing charged fragments are of the greatest interest because DNA molecules effectively interact with cationic compounds [29]. Adding tertiary amino groups into the structure of the macrocyclic platform promotes the reversible formation of cationic ammonium macrocyclic structures in aqueous solutions, which can effectively interact with DNA.

Examples of the introduction of thiaether fragments into the pillar[5]arene structure with high yields of target compounds [30,31,32,33] were previously demonstrated in our scientific group. Adding bulky thiaether fragments into the pillar[5]arene platform increases the lipophilicity of the target macrocyclic systems and eliminates the possibility of binding metal cations to the macrocyclic cavity [30]. An increase in the lipophilicity from cationic associates raises the degree of the adsorption of liposomes on the biofilm surface [34,35]. For instance, cationic lipids (such as stearylamine, dimethyldioctadecylammonium bromide and dimethylaminoethanecarbamoylcholesterol) were adsorbed in *St. Aureus* when incorporated into dipalmitoylphosphatidylcholine (DPPC) liposomes [35].

Thus, the creation of DNA-binding agents based on pillar[5]arene consists of the synthesis of a macrocyclic preorganized structure that has an affinity for DNA, but does not bind the metal cations.

To confirm the proposed hypothesis, potential DNA-binding macrocyclic agents based on pillar[5]arene were obtained by a stepwise synthesis from macrocycle **1**, which was synthesized according to the procedure in the literature [36]. To begin with, macrocycle **1** was converted into pillar[5]arene **2** at an 84% yield, according to our previously developed procedure [30] (Figure 1). We then obtained the amino derivatives of pillar[5]arene **3**–**6** at 60–89% yields by reacting macrocycle **2** with hydrazine hydrate, N,N-diethylaminoethylamine, N,N-dimethylaminoethylamine and morpholineethylamine (Figure 1).

The reactions were carried out for 40 h at a reflux in the solvent system of CH_3_OH:toluene = 1:1. The structure of macrocycles **1**–**6** was characterized by a complex of modern physical methods, including IR, ^1^H, ^13^C, ^1^H-^1^H NOESY NMR spectroscopy, MALDI mass spectrometry and elemental analysis data (see Appendix A, Appendix A).

A significant problem in the development of new polyfunctional drugs or delivery systems is their high affinity for metals [37]. This limits the penetration of the systems directly into the target cell [38] or prevents the targeted action on the biofilm matrix. Following this, in this work we showed the absence of an interaction of target macrocycles **3**–**6** with an excess of metal cations (Li^+^, Na^+^, K^+^, Cu^2+^, Ni^2+^, Mg^2+^, Fe^3+^, Ag^+^ and Co^2+^) (see Appendix A, Appendix A) by electron absorption spectroscopy. This fact expanded the potential of using the synthesized macrocycles in biological systems.

We chose salmon sperm DNA as a model to establish the fact of the interaction of macrocycles **3**–**6** with nucleic acids. The subsequent experiments were carried out in 50 mM of Tris-HCl buffer (pH = 6.5) in order to convert macrocycles **3**–**6** into a charged water-soluble ammonium form. Through the fermentation processes, the pH of the biofilm microenvironment was acidic (pH 4.5–6.5) [39]; therefore, the synthesized macrocycles should have been stable under standard biofilm conditions. Unfortunately, macrocycle **3** was poorly soluble in the proposed buffer; consequently, further studies of aggregation and complexation with salmon sperm DNA were carried out only with macrocycles **4**–**6** (Figure 1).

The ability of amido-amine derivatives **4**–**6** to interact with the model salmon sperm DNA in the Tris-HCl buffer (50 mM; pH = 6.5) was studied by UV-*vis* spectroscopy and dynamic light scattering (DLS). An important characteristic of the nanosystems used to combat pathogenic microorganisms is their size. The ideal size of nanoparticles to control the development of microbial biofilms is in the range of 5 nm to 300 nm; a size of 500 nm is extremely rarely allowed [40]. The next stage of the study was the evaluation of the aggregation characteristics of the obtained macrocycles **4**–**6** by DLS in the concentration range of 10^−4^–10^−6^ M in a buffer (pH = 6.5) (see Appendix A, Appendix A and Appendix A). Macrocycle **4** was characterized by the formation of self-associates at a concentration range of 1 × 10^−4^–1 × 10^−5^ M with a particle size of 350–370 nm and a polydispersity index (PDI) of 0.09–0.21 under the experimental conditions (Table 1). Meanwhile, macrocycle **6** at a concentration of 1 × 10^−5^ M formed submicron associates (d _average_ = 900 nm; PDI = 0.35) and pillararene **5** formed polydisperse systems over the entire range of concentrations (Table 1; Appendix A, Appendix A and Appendix A). Therefore, macrocycles **5** and **6** were not able to penetrate the biofilm matrix because of the large size of the self-associates. Only the nanoassociates of pillar[5]arene **4** containing fragments of (N,N-diethyl)aminoethylamide satisfied the required sizes (up to 500 nm).

The study of the aggregation characteristics of pillar[5]arene **4** with the model salmon sperm DNA was carried out at macrocycle:DNA _salmon sperm base pairs_ ratios of 2:1, 1:1, 1:2, 1:3, 1:4, 1:5, 1:6 and 1:10 in 50 mM Tris-HCl (pH = 6.5) by DLS (Table 1; Appendix A, Appendix A and Appendix A). Thus, we observed a polymodal distribution of particles with high values of the PDI in the DNA solutions in the concentration range of base pairs of 6 × 10^−4^–3 × 10^−5^ M. However, stable self-associates were formed with an average hydrodynamic diameter of 220 nm; the PDI = 0.24 and ζ-potential = 19.3 mV in **4** (1 × 10^−5^ M)/DNA _salmon sperm_ (6 × 10^−5^ M base pairs) of the system in 50 mM Tris-HCl (pH = 6.5) (Table 1; Figure 2A). The self-assembly of macrocycles **5** and **6** with salmon sperm DNA under the same experimental conditions was characterized by a similar behavior with the formation of particles with a size of 334 nm; the PDI = 0.30 and ζ-potential = 20 mV for the **5**/DNA _salmon sperm_ system. It was 345 nm in the case of the **6**/DNA _salmon sperm_ system (Table 1); the PDI = 0.42 and ζ-potential = 19.8 mV.

The aggregation properties of the synthesized compounds **4**–**6** were characterized by stable self-associates only in the case of macrocycle **4**, whose average hydrodynamic diameter decreased from 348 nm (self-associate **4**) to 220 nm (associate **4**/DNA) in the presence of a six-fold excess of model DNA. This fact confirmed the ability of the self-associates of **4** to penetrate the pathogenic biofilm and act as a DNA-binding agent. As a biofilm membrane surface is negatively charged, cationic compounds and carriers can damage the membrane integrity by interacting with it [41]. The ζ-potential data indicated a positively charged top layer of macrocycle **4**/DNA_salmon sperm_ associations, which promoted the interaction of the macrocycle with the surface of the pathogen matrix [40,41] and the release of its association with the DNA from the bacterial matrix. The obtained results of the aggregation processes corresponded with the necessary characteristics of the DNA-binding nanovector.

In order to establish the quantitative characteristics of the association processes of the obtained **4**/DNA associates, the interaction of salmon sperm DNA with macrocycle **4** was studied by UV-*vis* electron spectroscopy at pH = 6.5. The formation of associations between macrocycle **4** and the DNA was confirmed by spectrophotometric titration. The absorption spectra of the system in which the concentration of salmon sperm DNA (2 × 10^−5^–3 × 10^−4^ M base pairs) was varied at a constant concentration of macrocycle **4** (1 × 10^−5^ M) and showed a hyperchromic effect at the DNA wavelength (λ = 248 nm) (Figure 2B) (see Appendix A, Appendix A). The quantitative characteristics of the association process of macrocycle **4** with the DNA were established by spectrophotometric titration using the BindFit statistical model [42,43]. The data were consistent with the 1:2 (macrocycle **4**:DNA) binding model (see Appendix A, Appendix A). The association constants of pillararene **4** with DNA were lgK_ass1:1_ = 2.38 and lgK_ass1:2_ = 5.07 (see Appendix A, Appendix A). When host:guest = 1:1 and 2:1 binding models were used, the association constants were determined with a large error, which confirmed the correctness of our chosen 1:2 model (see Appendix A, Appendix A). Therefore, the **4**/DNA stoichiometry corresponded with the binding of one pillar[5]arene molecule to the phosphate fragments of two pairs of nucleic bases (Figure 3). This agreed with the literature data [44], which describes mainly the integration of decasubstituted amido-ammonium derivatives of pillar[5]arene into the small groove of DNA.

Pillar[n]arenes have been proven to be actively fluorescing molecules; on this basis, effective fluorescent sensors [45,46,47] and materials [48,49,50] have been developed. The obtained macrocycles **4**–**6** were also tested for their tendency to fluoresce in 50 mM Tris-HCl (pH = 6.5). Pillar [5]arenes **5** and **6** at a concentration of 1 × 10^−5^ M showed no fluorescent properties, in contrast to macrocycle **4**, which exhibited fluorescence with an emission maximum of λ = 324 nm (λ _ex_ = 280 nm) (Figure 2C). In the presence 60 μM base pairs of salmon sperm DNA, the emission of macrocycle **4** significantly decreased (Figure 2C). The hypochromic effect was accompanied by a slight hypochromic shift at 3 nm (λ_max_ = 321 nm), which confirmed the association of **4** with the DNA.

The next stage of the study was to evaluate the morphology of macrocycle **4** and **4**/DNA _salmon sperm_ complex associations by transmission electron microscopy (TEM). According to TEM (Figure 4), DNA _salmon sperm_ were irregularly shaped associates (Figure 4B); macrocycle **4** (Figure 4A) in water formed ordered aggregates with dendritic structures consisting of elliptical repeating units that were sized from 80–100 nm. However, the TEM image of **4**/DNA _salmon sperm_ associations (Figure 4C,D) showed spherical aggregates with an average diameter of 230 nm. Figure 4D reveals that a compacted DNA molecule was placed inside the spherical nanoassociate. This fact satisfied the application of macrocycle **4** as a DNA-binding agent of the extracellular DNA of the bacterial matrix, both in size characteristics (up to 500 nm) and in the morphology, with the possibility of capturing extracellular DNA and the subsequent inhibition of the development of a pathogenic biofilm.

Macrocyclic receptors capable of interacting with the extracellular DNA of pathogenic biofilms and insensitive to the presence of metal cations have previously been synthesized. However, these compounds can be highly toxic or, on the contrary, have no effect in vivo or in vitro on the development of pathogenic biofilms because of their highly organized and complex biochemical composition. As a result, we carried out experiments to assess the cytotoxicity of the synthesized compounds using the MTT test [51]. In order to comprehensively assess the effect of macrocycle **4** on the cell viability, the MTT test was performed with respect to both a cancerous model and normal cell lines. Widely used and experimentally studied human lung adenocarcinoma cells (A549) and bovine embryonic lung epithelial cells (LECs) were chosen as the cell lines. The experiment was accomplished by incubating the cell lines in the presence of macrocycle **4** for 24 h and a subsequent evaluation of the integral activity of cell dehydrogenases as an indicator of their viability. It was found that over the entire range of concentrations studied (0.5–100 μg/mL, corresponding with 1.7 × 10^−8^–3.4 × 10^−5^ M), macrocycle **4** did not statistically significantly reduce the viability of both the normal epithelial cells (LECs) and the A549 model cells (Figure 5A,B).

Due to the absence of cytotoxicity on the macrocyclic platform and the availability of the necessary pharmacophore fragments, the next step was to determine the antibacterial characteristics of pillararene **4** in relation to *Staphylococcus aureus* biofilms. The experiment revealed an inverse dependence of the effect of the pillararene **4** concentration on the suppression of the biofilm development; this reduced with an increase in the concentration with a maximum effect at a concentration of 10^−5^ M (Figure 5C and Appendix A, Appendix A). Thereby, macrocycle **4** at a concentration of 10^−5^ M was capable of reducing the thickness of the *St. Aureus* biofilm by 15% (Figure 5C). This fact could be explained by the ability of macrocycle **4** to form stable DNA-interacting self-associates at a concentration of 10^−5^ M (d_average_ = 350 nm, Table 1) and nanoscale aggregates of macrocycle **4** with salmon sperm DNA (d_average_ = 220 nm, Table 1).

An exceptional feature of the developed supramolecular system was the additional coordination center of pillar[5]arene in the form of a vacant cavity of the macrocycle [17,18,19,32]. The usage of pillar[5]arene **4** could aim to access design systems using the macrocyclic cavity, in which the macrocycle would play the role of a capsule molecule for various antibacterial drugs, suppressing the development of a resistance to antibiotics, controlling the solvate characteristics, improving the biocompatibility and prolonging their action. Therefore, this non-toxic supramolecular system could be used in combination therapies along with the antibiotic, enhancing its action [52].

To summarize, macrocycle **4** had the ability to self-associate and to form nanoassociates with DNA. It was a non-toxic DNA-binding agent and, in combination with antibiotics, could be the component enhancing the antibiotic action in the fight against bacterial infections.

## 4. Conclusions

This study proposed a new biocompatible supramolecular system based on a water-soluble derivative of pillar[5]arene **4** capable of inhibiting the development of *St. Aureus* biofilms. Its operating principle was based on the association of macrocycle **4** with DNA. The characteristics of the association of **4** with salmon sperm DNA were established by UV-*vis* spectroscopy. Macrocycle **4** interacted with the DNA base pairs at a ratio of 1:2 (macrocycle/DNA base pairs), which was further confirmed by fluorescence spectroscopy data. The association constants of **4** with salmon sperm DNA were lgK_ass1:1_ = 2.38 and lgK_ass1:2_ = 5.07, accordingly. The methods of dynamic light scattering and transmission electron microscopy showed that the interaction of pillar[5]arene containing thiaether and tertiary amino groups (10^−5^ M) with the model nucleic acid led to the formation of a stable nanosized association with an average particle size of 220 nm. The obtained compounds did not exhibit any pronounced toxicity to human lung adenocarcinoma cells (A549) and bovine embryonic lung epithelial cells (LECs). A hypothesis about the possible usage of the synthesized macrocycle as a DNA-binding agent in a biofilm matrix was confirmed using the example of *St. Aureus*. It was found that pillar[5]arene **4** at a concentration of 10^−5^ M was able to reduce the thickness of the *St. Aureus* biofilm by 15%. Due to the situation that has developed in recent years from the formation of a resistance of pathogenic microorganisms to the action of antibiotics, the usage of synthetic non-toxic DNA-binding agents increases the effectiveness of combined therapies for infectious diseases.

## Data Availability

The data presented in this study are available in the Appendix A.

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
