# Peer review of "Toward Pathogenic Biofilm Suppressors: Synthesis of Amino Derivatives of Pillar[5]arene and Supramolecular Assembly with DNA"

_pharmaceutics, 2023, doi:10.3390/pharmaceutics15020476_

Round 1
Reviewer 1 Report
1. Summary
The manuscript entitled, as “Toward Pathogenic Biofilm Suppressors: Synthesis of Amino 2 Derivatives of Pillar[5]arenes and Supramolecular Assembly with DNA”, is a short article describes new amino-derivatives of pillar[5]arene is biocompatible supramolecular system for inhibiting the growth of bacterial biofilm and future use of the compound to provide treatment options for microorganism’s resistance to the action of antibiotic in a safe manner without any cytotoxicity. Author(s) have synthesized new macrocyclic compound 4-6 which have less affinity to metal ions, forms complex with DNA with a diameter of 220 nm as measured by UV-vis and DLS methods. This study highlighted the newly synthesized macrocyclic compound 4 was able to reduce the Staphylococcus aureus biofilm by 15% which have therapeutic potential. The manuscript looks great in current format I would see some improvement in the manuscript. I would suggest following things for the authors to improve the manuscript.
2. General comments
A. The manuscript was written well and looks great. I was able to follow the entire manuscript easily from Abstract to conclusion.
B. In results section- from starting point it looks that it’s not just results section, it have discussion part in it- I would recommend the authors to change the heading from Results to Results and Discussion section.
C. Figure S30 and Table S3 where authors describe the concentration effect of macrocycle 4 on Staphylococcus aureus should be shifted to main figures- I think it is important results of study and to know biological relevance of the compound.bAlso, should add error bars to the figure S30 because 15% is not that big value difference- it may be falling in the error limit.
D. Figure 2C where authors describe reduce in fluorescent intensity of the compound upon bound to DNA. I feel that when you added DNA solution to the fluorescent compound it may dilute the concentration of the compound in the solution that dilutes fluorescence intensity value of the original compound- I would recommend authors to rule out the possibility of the dilution effect on fluorescence- provide proper control experiment or explained to the results.
E. Image quality of the figure is not at all good. They were blur and it is hard to read the labelling on the figure. I would suggest the authors to upload high quality images.
With the above changes, I would accept the manuscript in this journal.
Author Response
- « The manuscript was written well and looks great. I was able to follow the entire manuscript easily from Abstract to conclusion»
Answer: We thank the reviewer for the good assessment of our study!
- « In results section- from starting point it looks that it’s not just results section, it have discussion part in it- I would recommend the authors to change the heading from Results to Results and Discussion section»
Answer: The heading change proposed by the reviewer was made and included in the text of the article.
- « Figure S30 and Table S3 where authors describe the concentration effect of macrocycle 4 on Staphylococcus aureus should be shifted to main figures- I think it is important results of study and to know biological relevance of the compound. Also, should add error bars to the figure S30 because 15% is not that big value difference- it may be falling in the error limit»
Answer: Figure S30 has been moved from the ESI file to the main article and added as Figure 5c. Mean of grouped data and population confidence intervals (p≤0.05) were calculated. The significance of differences in group data compared with the option without pillar[5]arene treatment was established by the nonparametric Mann-Whitney U test. It was decided to leave the table with the experimental data in the ESI file, since the updated Figure 5c fully corresponds to the quality of the experimental data.
Figure 5. Survival of LEC (a), A 549 cells (b) after incubation with pillar[5]arene 4 during 24 hours; (с) Effect of pretreatment of the adhesive surface of the plate with pillar[5]arene 4 solution on the biofilm formation ability of Staphylococcus aureus.
- « Figure 2C where authors describe reduce in fluorescent intensity of the compound upon bound to DNA. I feel that when you added DNA solution to the fluorescent compound it may dilute the concentration of the compound in the solution that dilutes fluorescence intensity value of the original compound- I would recommend authors to rule out the possibility of the dilution effect on fluorescence- provide proper control experiment or explained to the results»
Answer: The standard experimental procedure described in ESI was used for studying the fluorescence characteristics of individual compounds (macrocycles 2, 3, 4 and DNA) and 4/DNA complex. Additional dilution during the experiment and the formation of the complex does not occur; consequently, strong fluorescence quenching reflects only the formation of the complex between macrocycle 4 and DNA. The concentration of macrocycle 4 was fixed (10-5 M). For clarity, the text of the article and the ESI have been amended.
A fragment was added to the manuscript: In the presence of 60 µM base pairs of salmon sperm DNA the emission of macrocycle 4 significantly decreased (Fig. 2c).
- « Image quality of the figure is not at all good. They were blur and it is hard to read the labelling on the figure. I would suggest the authors to upload high quality images »
Answer: Unfortunately, converting the article file from the original .docx format to .pdf significantly degrades the quality of the figures. All images offered in the article are downloadable in the originally proper quality and suitable for scientific publications.
We have tried to enhance image quality to the highest possible in the edited version of the manuscript.

Reviewer 2 Report
Dear authors,
the manuscript you proposed describes a new biocompatible supramolecular system based on a water-soluble derivative of pillar[5]arene 4 capable of inhibiting the development of St. Aureus biofilms. The manuscript is interesting, but I have following comments:
1) Why did the authors decide to divide the materials and methods? I suggest putting them together in the main article and not in the supporting.
2) In figure 1, I suggest putting the scale in the image and say with which technique/equipment they captured the image.
3) Figure 4, I suggest the authors to change the 4D figure with another TEM image, because in this way it seems that the authors have observed only one area of the sample.
4) Cell viability is not supported by a population statistic (e.g., p value). Furthermore, for the sake of completeness, I suggest also insert a morphological data to understand if even morphologically the cells do not undergo any changes.
Author Response
- «Why did the authors decide to divide the materials and methods? I suggest putting them together in the main article and not in the supporting»
Answer: We use standard methods for conducting experiments, which in the text of the main article may be perceived by the MDPI Pharmaceutics editors as plagiarism. Therefore, it was decided to transfer the data on the instrumental base and research methods to the supporting information (the ESI file), leaving only the data of the synthesis of new compounds in the text of the main article. In case if it necessary to combine materials and methods in the main article, then we’ll agree with the opinion of the reviewer and will be ready to combine this material in the text of the main article.
- «In figure 1, I suggest putting the scale in the image and say with which technique/equipment they captured the image»
Answer: The scale in Figure 1 is added to the TEM image. The technical characteristics of the equipment are described in the ESI file in section 1. Materials and methods.
The image has been replaced in the manuscript:
Figure 1. The sketch presents the concept of using pillar[5]arenes as DNA-binding agents to suppress the development of pathogenic biofilms.
- «Figure 4, I suggest the authors to change the 4D figure with another TEM image, because in this way it seems that the authors have observed only one area of the sample»
Answer: The Figure 4D has been replaced with a different TEM image of the complex 4/DNA.
Figure 4. TEM images: (a) macrocycle 4 (1×10-5 M); (b) DNA salmon sperm (6×10-5 M); (c), (d) system 4 (1×10-5 M) / DNA salmon sperm (6×10-5 M base pairs).
- « Cell viability is not supported by a population statistic (e.g., p value). Furthermore, for the sake of completeness, I suggest also insert a morphological data to understand if even morphologically the cells do not undergo any changes»
Answer:
Figure S30 has been removed from the ESI file and added to the main article as Figure 5c. Mean of grouped data and population confidence intervals (p≤0.05) were calculated. The significance of differences in group data compared with the option without pillar[5]arene treatment was established using the nonparametric Mann-Whitney U test. This work is the first study where macrocyclic compounds based on pillar[5]arenes are presented as non-toxic DNA-binding agents. First of all, this was an exploratory study to evaluate the efficiency of the usage of pillar[5]arenes as DNA-binding agents. At that point, we chose the most typical pathogenic microorganism S. Aureus and evaluated the effect of the synthesized pillar[5]arene to the development of its biofilm. Undoubtedly, our further work will be aimed to study the morphological features of cells treated with solutions of macrocyclic systems.
